# A Novel Unified Parametric Assumption for Nonconvex Optimization

## Abstract

Nonconvex optimization is central to modern machine learning, but the general framework of nonconvex optimization yields weak convergence guarantees that are too pessimistic compared to practice. On the other hand, while convexity enables efficient optimization, it is of limited applicability to many practical problems. To bridge this gap and better understand the practical success of optimization algorithms in nonconvex settings, we introduce a novel unified parametric assumption. Our assumption is general enough to encompass a broad class of nonconvex functions while also being specific enough to enable the derivation of a unified convergence theorem for gradient-based methods. Notably, by tuning the parameters of our assumption, we demonstrate its versatility in recovering several existing function classes as special cases and in identifying functions amenable to efficient optimization. We derive our convergence theorem for both deterministic and stochastic optimization, and conduct experiments to verify that our assumption can hold practically over optimization trajectories.

## 1 Introduction

There is a large disconnect between the theory and practice of nonconvex optimization with first-order methods. The theory for nonconvex optimization allows us only to guarantee convergence to a stationary point, or at most, a higher-order stationary point (Carmon et al., 2017a;b). In practice, neural scaling laws show smooth decreases in the loss function value as the number of training steps increases (Kaplan et al., 2020). In contrast, convex optimization theory typically allows us to derive tight guarantees on the function value (Nesterov, 2018), but is too restrictive to apply to nonconvex models directly. This discrepancy has motivated researchers to develop intermediate theoretical frameworks that allow us to obtain stronger convergence guarantees without losing too much applicability. These developments include star convexity (Nesterov & Polyak, 2006), quasi-convexity (Hardt et al., 2018; Bu & Mesbahi, 2020), the Polyak-Łojasiewicz (PL) condition (Polyak, 1963; Liu et al., 2021), Aiming (Liu et al., 2023), and the $\alpha$-$\beta$ conditions (Islamov et al., 2024).

**Problem statement.** We are primarily concerned with the minimization problem

$$\min_{x \in \mathbb{R}^d} f(x),$$

where $f(x) : \mathbb{R}^d \to \mathbb{R}$ is a differentiable function. We study gradient descent variants of the form

$$x^{k+1} = x^k - \gamma^k \nabla f\left(x^k\right),$$

where $\gamma^k > 0$ is a stepsize and $\nabla f(x^k)$ represents the gradient of the function $f$ at the current point $x^k$. Our analysis also extends to stochastic gradient descent (Section 2.5).

**A novel unified assumption.** We build on this line of work by introducing a new assumption that allows us to obtain convergence guarantees for nonconvex optimization. Our unified framework is broadly applicable– it subsumes prior assumptions on nonconvex optimization and allows for unified analysis of convex and nonconvex objectives. The main idea of our framework is that even in complex nonconvex landscapes, effective optimization algorithms rely on the gradient possessing a degree of directional alignment towards the set of solutions. To formalize this, we first make the assumption that a set of solutions exists.

**Assumption 1.** The function $f$ is continuously differentiable and has a nonempty set $S \subseteq \mathbb{R}^d$ of global minimizers. Let $f^\star$ denote the minimum value of the function $f$.

We now introduce our main assumption, an inequality that relates the gradient at any point $x$ to its projection onto a subset $\tilde{S}$ of optimal solutions, using a progress function $P(x; \tilde{S})$ to quantify proximity to this set.

**Assumption 2.** There exists constants $c_1 > 0$ and $c_2 \geq 0$ such that for any $x \in \mathbb{R}^d$ one can find a projection point $x_p \in \arg\min_{y \in \tilde{S}} \|x - y\|^2$ satisfying

$$\langle \nabla f(x), x - x_p \rangle \geq c_1 P(x; \tilde{S}) - c_2 I_X(x),$$

where $\tilde{S} \subseteq S$, $S \subseteq \mathbb{R}^d$ is a set of global minimizers of $f$, $\tilde{S} \neq \emptyset$, $P(x; \tilde{S})$ is a nonnegative function of the argument $x \in \mathbb{R}^d$, $X \subseteq \mathbb{R}^d$, $I_X(x) := \begin{cases} 1, & \text{if } x \in X \\ 0, & \text{otherwise} \end{cases}$.

Assumption 2 has a clear and intuitive interpretation: the progress function controls how "informative" the gradient is in pointing us towards the set of minimizers, while the constants $c_1$ and $c_2$ control how stringent this information is.

**Our contributions.** We develop a new framework for analyzing gradient descent under Assumption 2. We demonstrate that our framework recovers classical convergence guarantees for convex optimization as a special case, and also subsumes several existing assumptions in nonconvex optimization (such as quasiconvexity and the aiming condition). We provide convergence analysis under this new assumption for gradient descent (Theorem 2.1) and stochastic gradient descent (Theorem 2.5) and demonstrate the flexibility of these theorems in deriving new convergence guarantees. Finally, we provide experimental validation for how applicable our assumption is in half-space learning with the sigmoid, training MLPs on Fashion-MNIST, and training the ResNet model on CIFAR-10.

## 1.1 BRIEF LITERATURE REVIEW

Bridging the gap between theory and practice in nonconvex optimization has spurred significant research into developing more refined analytical frameworks. Classical convex optimization theory provides strong convergence guarantees, but its assumptions are often too restrictive for modern machine learning. Conversely, standard nonconvex optimization results guarantee convergence to stationary points and do not reflect the empirical success of first-order methods in deep learning. Convexity can be seen as controlling the lower curvature of a function while smoothness controls the upper curvature. The literature has explored generalizations and alternatives to both.

**Alternatives to convexity.** The Polyak-Łojasiewicz (PL) condition (Polyak, 1963; Liu et al., 2021) is a prominent example that relates the function value to the gradient norm, provides a lower bound on the function growth, and ensures linear convergence under certain conditions. Quasi-convexity (Hardt et al., 2018) and star-convexity (Nesterov & Polyak, 2006) represent other relaxations of convexity that have been studied in optimization. More recently, conditions like the Aiming property (Liu et al., 2023) and the $\alpha$-$\beta$ conditions (Islamov et al., 2024) have emerged as tools to characterize the loss landscapes of neural networks and analyze the convergence of optimization algorithms.

**Alternatives to smoothness.** Recent work has explored alternatives to smoothness that may more accurately describe neural network optimization, e.g. generalized smoothness (Zhang et al., 2020a; Xie et al., 2024), directional sharpness or smoothness (Pan & Li, 2022; Mishkin et al., 2024), and local smoothness (Berahas et al., 2023).

**Assumptions on the stochastic gradients.** Another line of work has considered the various properties of the stochastic gradient noise, and its effect on the convergence of gradient-based methods, see e.g. (Khaled & Richtárik, 2020; Faw et al., 2022; Zhang et al., 2020b). Our work is primarily aimed at relaxing convexity and is therefore orthogonal to these results.

## 2 MAIN THEORY & RESULTS

In this section, we first discuss further Assumption 2 and its implications, then present our convergence theory for gradient descent under this assumption, followed by stochastic gradient descent.

## 2.1 DISCUSSION OF ASSUMPTION 2

To analyze Assumption 2, we start by considering the simpler setting $c_2 = 0$. In this case, Assumption 2 takes the form

$$\langle \nabla f(x), x - x_p \rangle \geq c_1 P(x; \tilde{S}) \geq 0 \quad \text{for all } x \in \mathbb{R}^d.$$

This means that the negative gradient $-\nabla f(x)$ points toward $\tilde{S}$ in the sense that $-\nabla f(x)$ is nontrivially correlated with the direction $x_p - x$. The term $c_1 P(x; \tilde{S})$ can tighten or relax this correlation depending on the choices of $c_1$ and $P(x; \tilde{S})$, leading to narrower or wider classes of functions. Introducing $c_2$ relaxes the correlation, possibly allowing the inner product to be negative at certain points $x \in X$.

Now, consider the case where $x \in \mathbb{R}^d$ is a stationary point of $f$, i.e., $\nabla f(x) = 0$. From Assumption 2 we have that $P(x; \tilde{S}) \leq \frac{c_2}{c_1}$. This implies, in terms of the measure $P(x; \tilde{S})$, the stationary point $x$ is not too far from the set $\tilde{S}$.

A specific and natural choice for the progress function in Assumption 2 is $P(x; \tilde{S}) = f(x) - f^\star$, with $c_1 = 1$, $c_2 = 0$, and $\tilde{S} = \{x^\star\}$, where $x^\star \in S$. With these choices, Assumption 2 becomes

$$\langle \nabla f(x), x - x^\star \rangle \geq f(x) - f^\star \quad \text{for all } x \in \mathbb{R}^d,$$

which is a simple consequence of the convexity of $f$ from the standard convex analysis.

For additional examples of various classes of functions derived from Assumption 2 that yield meaningful convergence results, please refer to Section 2.4 and 2.5, where by adjusting the parameters of Assumption 2, we can recover many well-known function classes as special cases, including convex, strongly convex, weak quasi-convex functions (Hardt et al., 2018), strongly weak quasi-convex functions (Bu & Mesbahi, 2020), and functions satisfying the Aiming (Liu et al., 2023) property or the $\alpha$-$\beta$ condition (Islamov et al., 2024). Moreover, this framework also reveals entirely new classes.

**Role of the parameters** $(c_1, c_2, \tilde{S})$. Now, let us examine how the flexibility of the choices $(c_1, c_2, \tilde{S})$ in Assumption 2 leads to wider classes of functions for the particular choice of $P(x, \tilde{S}) = f(x) - f^\star$, where we assume without loss of generality that $f^\star = 0$. We include examples of how our assumption subsumes existing conditions and allow for relaxed ones in Table 1, including different examples of functions $f(x)$, $x \in \mathbb{R}$ (see Figure 4 and Section A for details):

$$f_1 = x^2, \qquad f_2 = \begin{cases} f_1, & x \geq -1 \\ 4\sqrt{-x} - 3, & x < -1 \end{cases}, \qquad f_3 = \frac{x^4}{2} - x^2 + \frac{1}{2},$$

$$f_4 = x^4 - \frac{10}{3}x^3 + 3x^2, \qquad f_5 = \begin{cases} f_4, & x \geq 0 \\ f_2, & x < 0 \end{cases}.$$

Table 1: Examples of function classes described by Assumption 2 for $P(x, \tilde{S}) = f(x) - f^\star$, $f^\star = 0$, $\tilde{S} \subseteq S$, and different $(c_1, c_2, \tilde{S})$. Here, $S \subseteq \mathbb{R}$ is a set of global minimizers of $f$, $x^\star \in S$.

| $c_1, \tilde{S}$ | $c_2 = 0$ | $c_2 \geq 0$ |
|---|:---:|:---:|
| $c_1 = 1, \tilde{S} = \{x^\star\}$ | CONSEQUENCE OF CONVEXITY | NEW |
| EXAMPLES: | $f_1$ | $f_1, f_3, f_4$ |
| $c_1 > 0, \tilde{S} = \{x^\star\}$ | WEAK QUASI-CONVEXITY (HARDT ET AL., 2018) | NEW |
| EXAMPLES: | $f_1, f_2$ | $f_1, f_2, f_3, f_4, f_5$ |
| $c_1 > 0, \tilde{S} = S$ | AIMING CONDITION (LIU ET AL., 2023) | NEW |
| EXAMPLES: | $f_1, f_2$ | $f_1, f_2, f_3, f_4, f_5$ |

We can observe that incorporating constants into Assumption 2, allowing $c_1 \neq 1$ and $c_2 \neq 0$, leads to broader classes of functions. When $c_2 \neq 0$, Assumption 2 can describe functions with local minima

and saddle points. Note that for certain functions, choosing $c_1 \neq 1$ and $\tilde{S} = S$ allows us to satisfy Assumption 2 with a smaller constant $c_2$:

- Specifically, for a fixed $c_1 = 1$, if $\tilde{S} = \{x^\star\}$, $x^\star = 1$, then $f_3$ satisfies Assumption 2 with $c_2 \approx 1.437$, and if $\tilde{S} = S$, then $f_3$ satisfies Assumption 2 with $c_2 = 0.5$. In both of these cases, we choose the smallest $c_2$ for the given $c_1$.

- For the function $f_4$, it can be shown that $f_4$ satisfies Assumption 2 with $c_1 = 1$, $c_2 \approx 1.013$, or $c_1 = 0.1$, $c_2 \approx 0.467$. Also, if $f_3$ is considered with $\tilde{S} = S$, it satisfies Assumption 2 with $c_1 = 1$, $c_2 \approx 0.5$, or $c_1 = 0.1$, $c_2 = 0.05$. In all these examples, we select the smallest $c_2$ for the given $c_1$.

## 2.2 Main Convergence Theorem

In this section, we examine the convergence guarantees we can obtain under the proposed Assumption 2.

**Theorem 2.1.** *Let Assumptions 1 and 2 be satisfied. Further assume that the stepsize $\gamma^k$ satisfies the relations*

$$0 < \gamma^k \leq (2 - \alpha) \frac{\langle \nabla f(x^k), x^k - x_p^k \rangle + c_2 I_X(x^k) + \beta^k}{\|\nabla f(x^k)\|^2}$$

*that holds for all $k \geq 0$, where $0 < \alpha < 2$, $\beta^k \geq 0$. Then we have the following "descent inequality" that holds for all $k \geq 0$ and the convergence result:*

$$\|x^{k+1} - x_p^{k+1}\|^2 \leq \|x^k - x_p^k\|^2 - \alpha c_1 \gamma^k P(x^k; \tilde{S}) + (2 - \alpha)\beta^k \gamma^k + 2c_2 \gamma^k I_X(x^k),$$

$$\min_{k \in \{0,...,K\}} P(x^k; \tilde{S}) \leq \frac{\sum_{k=0}^K \gamma^k P(x^k; \tilde{S})}{\sum_{k=0}^K \gamma^k} \leq \frac{\|x^0 - x_p^0\|^2}{\alpha c_1 \sum_{k=0}^K \gamma^k} + C^K,$$

*where $C^K := \frac{\sum_{k=0}^K \gamma^k (2-\alpha)\beta^k}{\alpha c_1 \sum_{k=0}^K \gamma^k} + \frac{2c_2 \sum_{k=0}^K \gamma^k I_X(x^k)}{\alpha c_1 \sum_{k=0}^K \gamma^k}$.*

**Analysis of Theorem 2.1.** Theorem 2.1 provides convergence guarantees for $P(x; \tilde{S})$ within a neighborhood $C^K$, given that the sum of the stepsizes, $\sum_{k=0}^K \gamma^k$, is sufficiently large. However, achieving a precise convergence rate to the neighborhood requires additional assumptions on $P(x; \tilde{S})$, the function $f$, or the stepsizes $\gamma^k$. For instance, under assumptions such as $P(x; \tilde{S}) = f(x) - f^\star$ and smoothness (Corollary 2.2), bounded gradients (Corollary 2.3), or decreasing stepsizes (Corollary 2.4), we can establish convergence to a neighborhood within the framework of Theorem 2.1. However, the latter two results—bounded gradients and decreasing stepsizes—are only meaningful if $P(x; \tilde{S})$ satisfies certain regularity properties, which are discussed in detail in Section 2.3. If we further assume that only a small finite number of points $x^k$ belong to $X$, such that $\frac{\sum_{k=0}^K \gamma^k I_X(x^k)}{\sum_{k=0}^K \gamma^k} = \mathcal{O}\left(\frac{1}{K^\theta}\right)$, $\theta > 0$, then the convergence neighborhood can shrink. In Section 3, we empirically observe that along the training trajectories, the set $X$ contains only a few "problematic" points.

**Corollary 2.2.** *Under the assumptions of Theorem 2.1 with $P(x; \tilde{S}) = f(x) - f^\star$, if we additionally assume that $f$ is L-smooth, and choose $\gamma^k = \frac{c_1(f(x^k)-f^\star)}{\|\nabla f(x^k)\|^2}$, $\alpha = 1$, $\beta^k = 0$, then we obtain*

$$\min_{k \in \{0,...,K\}} f(x^k) - f^\star \leq \frac{2L \|x^0 - x_p^0\|^2}{c_1^2(K+1)} + C^K,$$

*where $C^K := \frac{2c_2 \sum_{k=0}^K \gamma^k I_X(x^k)}{c_1 \sum_{k=0}^K \gamma^k} \leq \frac{4Lc_2 \sum_{k=0}^K \gamma^k I_X(x^k)}{c_1^2(K+1)}$.*

Note that for $\tilde{S} = \{x^\star\}$, $x^\star \in S$, $c_1 = 1$, $c_2 = 0$, Corollary 2.2 presents a well-known result from standard convex analysis for the Polyak stepsize (Polyak, 1987).

**Corollary 2.3.** *Under the assumptions of Theorem 2.1, if we additionally assume that $f$ has bounded gradients, i.e., $\|\nabla f(x)\| \leq G$ for all $x \in \mathbb{R}^d$, and choose $\gamma^k = (2-\alpha)\frac{c_1 P(x^k;\tilde{S})+\beta^k}{\|\nabla f(x^k)\|^2}$, then we obtain*

$$\min_{k\in\{0,\dots,K\}} P(x^k;\tilde{S}) \leq \frac{G\|x^0 - x_p^0\|}{\sqrt{(2-\alpha)\alpha c_1}} \frac{1}{\sqrt{K+1}} + C^K,$$

*where $C^K := \frac{\sum_{k=0}^K \gamma^k(2-\alpha)\beta^k}{\alpha c_1 \sum_{k=0}^K \gamma^k} + \frac{2c_2 \sum_{k=0}^K \gamma^k I_X(x^k)}{\alpha c_1 \sum_{k=0}^K \gamma^k}$.*

**Corollary 2.4.** *Under the assumptions of Theorem 2.1, if we additionally assume that $\gamma^k \leq \gamma^{k-1}$ for $k = 1, \dots, K$, then we obtain*

$$\min_{k\in\{0,\dots,K\}} P(x^k;\tilde{S}) \leq \frac{D_{\max}^2}{\alpha c_1 \gamma^K (K+1)} + \tilde{C}^K,$$

*where $\tilde{C}^K := \frac{(2-\alpha)\sum_{k=0}^K \beta^k + 2c_2 \sum_{k=0}^K I_X(x^k)}{\alpha c_1 (K+1)}$, $D_{\max}^2 := \max_{k\in\{0,\dots,K\}} \|x^k - x_p^k\|^2$.*

## 2.3 SPECIAL CASES

Let us consider some examples of stepsizes that satisfy Theorem 2.1 for a specific choice of $P(x;\tilde{S}) = f(x) - f^\star$ and $\tilde{S} = \{x^\star\}$, $x^\star \in S$. These results are summarized in Table 2. From the table, we observe that for various stepsizes of the Polyak type (Polyak, 1987; Loizou et al., 2021; Orvieto et al., 2022), convergence is achieved up to a neighborhood under the assumptions of Theorem 2.1, along with additional conditions such as the smoothness of the function $f$ or the boundedness of its gradients, i.e., $\|\nabla f(x)\| \leq G$ for all $x \in \mathbb{R}^d$ (see Section C for details).

Table 2: Examples of stepsizes that satisfy Theorem 2.1 for $\alpha = 1$, $P(x,\tilde{S}) = f(x) - f^\star$, $\beta^k = 0$, where $\tilde{S} = \{x^\star\}$, $x^\star \in S$. Here, $l^\star \leq f^\star$, $c^k = \sqrt{k+1}$, $\sigma^2 := f^\star - l^\star$.

| STEPSIZE, $\gamma^k$ | EXTRA ASSUMPTION | CONVERGENCE RATE |
|---|---|---|
| $\dfrac{c_1(f(x^k)-f^\star)}{\|\nabla f(x^k)\|^2}$ | SMOOTHNESS | $\mathcal{O}\left(\frac{1}{K}\right) + C^K$ |
| | BOUNDED $\nabla f$ | $\mathcal{O}\left(\frac{1}{\sqrt{K}}\right) + C^K$ |
| $\dfrac{c_1(f(x^k)-l^\star)}{\|\nabla f(x^k)\|^2}$ | SMOOTHNESS | $\mathcal{O}\left(\frac{1}{K}\right) + C^K + \sigma^2$ |
| | BOUNDED $\nabla f$ | $\mathcal{O}\left(\frac{1}{\sqrt{K}}\right) + C^K + \sigma^2$ |
| $\min\left\{\dfrac{c_1(f(x^k)-l^\star)}{c^k\|\nabla f(x^k)\|^2}, \dfrac{\gamma^{k-1}c^{k-1}}{c^k}\right\}$ | SMOOTHNESS | $\mathcal{O}\left(\frac{1}{\sqrt{K}}\right) + C^K$ |

## 2.4 EXAMPLES OF FUNCTION CLASSES

Next, let us consider some choices of $P(x;\tilde{S})$, $c_1$, and $c_2$ in Assumption 2 that describe specific classes of functions and lead to meaningful convergence results. Our first example is one we have already mentioned before.

***Example 1.*** Let $P(x;\tilde{S}) = f(x) - f^\star$.

Note that if $\tilde{S} = \{x^\star\}$, $x^\star \in S$, $c_2 = 0$, then Assumption 2 is equivalent to the definition of $c_1$-weak quasi-convex functions (Hardt et al., 2018). If additionally $c_1 = 1$, then Assumption 2 follows from the convexity of the function $f$.

Consider using the Polyak stepsize $\gamma^k = \frac{c_1 P(x^k;\tilde{S})}{\|\nabla f(x^k)\|^2} = \frac{c_1(f(x^k)-f^\star)}{\|\nabla f(x^k)\|^2}$, with $\alpha = 1$, and $\beta^k = 0$. If we additionally assume that $f$ is $L$-smooth, then $\gamma^k \geq \frac{c_1}{2L}$ and from Corollary 2.2 we get the following

convergence result

$$\min_{k \in \{0,\ldots,K\}} f(x^k) - f^\star \leq \frac{2L \left\| x^0 - x_p^0 \right\|^2}{c_1^2 (K+1)} + \frac{2c_2 \sum_{k=0}^{K} \gamma^k I_X(x^k)}{c_1 \sum_{k=0}^{K} \gamma^k}.$$

If $c_2 = 0$ or $\sum_{k=0}^{K} \gamma^k I_X(x^k) = \mathcal{O}(1)$, then we obtain an $\mathcal{O}\left(\frac{1}{K}\right)$ convergence rate for $\min_{k \in \{0,\ldots,K\}} f(x^k) - f^\star$ under Assumptions 1, 2, and the smoothness of $f$.

If, instead of the smoothness of $f$, we assume that $f$ has bounded gradients, then from Corollary 2.3, we get the following convergence result

$$\min_{k \in \{0,\ldots,K\}} f(x^k) - f^\star \leq \frac{G \left\| x^0 - x_p^0 \right\|}{c_1 \sqrt{K+1}} + \frac{2c_2 \sum_{k=0}^{K} \gamma^k I_X(x^k)}{c_1 \sum_{k=0}^{K} \gamma^k}.$$

If $c_2 = 0$, then we obtain an $\mathcal{O}\left(\frac{1}{\sqrt{K}}\right)$ convergence rate for $\min_{k \in \{0,\ldots,K\}} f(x^k) - f^\star$ under Assumptions 1, 2, and the boundedness of the gradients of $f$.

***Example 2.*** Let $P(x; \tilde{S}) = \frac{1}{L} \|\nabla f(x)\|^2$, $L > 0$.

Note that if $\tilde{S} = \{x^\star\}$, $x^\star \in S$, $c_1 = 1$, $c_2 = 0$, then Assumption 2 follows from the convexity and $L$-smoothness of the function $f$. Here, we used the fact that $f$ is $L$-smooth and convex , which is equivalent to the property that for all $x, y \in \mathbb{R}^d$

$$\frac{1}{L} \|\nabla f(x) - \nabla f(y)\|^2 \leq \langle \nabla f(x) - \nabla f(y), x - y \rangle.$$

Let us choose $\gamma^k = \frac{c_1 P(x^k; \tilde{S})}{\|\nabla f(x^k)\|^2} = \frac{c_1}{L}$. Then, by setting $\alpha = 1$, $\beta^k = 0$, $\gamma^k$ satisfies the relations of Theorem 2.1. Therefore, we get the following convergence result

$$\min_{k \in \{0,\ldots,K\}} \|\nabla f(x^k)\|^2 \leq \frac{L^2 \left\| x^0 - x_p^0 \right\|^2}{c_1^2 (K+1)} + \frac{2c_2 \sum_{k=0}^{K} \gamma^k I_X(x^k)}{c_1 \sum_{k=0}^{K} \gamma^k}.$$

If $c_2 = 0$ or $\sum_{k=0}^{K} \gamma^k I_X(x^k) = \mathcal{O}(1)$, then we obtain an $\mathcal{O}\left(\frac{1}{K}\right)$ convergence rate for $\min_{k \in \{0,\ldots,K\}} \|\nabla f(x^k)\|^2$ under Assumptions 1, 2.

***Example 3.*** Let $P(x; \tilde{S}) = f(x)$, $f^\star = 0$.

Note that if $\tilde{S} = S$, where $S$ is a nonempty set, $c_2 = 0$, then Assumption 2 is equivalent to the Aiming condition (Liu et al., 2023).

Let us choose $\gamma^k = \frac{c_1 P(x^k; \tilde{S})}{\|\nabla f(x^k)\|^2} = \frac{c_1 f(x^k)}{\|\nabla f(x^k)\|^2}$. Then, by setting $\alpha = 1$, $\beta^k = 0$, $\gamma^k$ satisfies the relations of Theorem 2.1. Similar to the previous examples, assuming that $f$ is $L$-smooth, we can show that

$$\min_{k \in \{0,\ldots,K\}} f(x^k) \leq \frac{2L \left\| x^0 - x_p^0 \right\|^2}{c_1^2 (K+1)} + \frac{2c_2 \sum_{k=0}^{K} \gamma^k I_X(x^k)}{c_1 \sum_{k=0}^{K} \gamma^k},$$

and assuming that $f$ has bounded gradients, we get

$$\min_{k \in \{0,\ldots,K\}} f(x^k) \leq \frac{G \left\| x^0 - x_p^0 \right\|}{c_1 \sqrt{K+1}} + \frac{2c_2 \sum_{k=0}^{K} \gamma^k I_X(x^k)}{c_1 \sum_{k=0}^{K} \gamma^k}.$$

If $c_2 = 0$ or $\sum_{k=0}^{K} \gamma^k I_X(x^k) = \mathcal{O}(1)$, then we obtain an $\mathcal{O}\left(\frac{1}{K}\right)$ convergence rate for $\min_{k \in \{0,\ldots,K\}} f(x^k)$ under Assumptions 1, 2, and the smoothness of $f$. Also, if $c_2 = 0$, then we get an $\mathcal{O}\left(\frac{1}{\sqrt{K}}\right)$ convergence rate for $\min_{k \in \{0,\ldots,K\}} f(x^k)$ under Assumptions 1, 2, and the boundedness of the gradients of $f$.

Additional examples of function classes are provided in Section D.

## 2.5 EXTENSION TO THE STOCHASTIC SETTING

**Problem formulation.** In this subsection, we extend our results to the stochastic optimization problem

$$\min_{x \in \mathbb{R}^d} \{f(x) := \mathrm{E}_{\xi \sim \mathcal{D}} [f_\xi(x)]\},$$

where $\xi$ are samples from some distribution $\mathcal{D}$. We solve it using the stochastic gradient method

$$x^{k+1} = x^k - \gamma^k \nabla f_{\xi^k}(x^k).$$

**Assumptions.** To facilitate our convergence analysis, we make the following assumption on $f_\xi$.

**Assumption 3.** There exists constants $c_{1\xi} > 0$ and $c_{2\xi} \geq 0$ such that for any $x \in \mathbb{R}^d$ one can find a projection point $x_p \in \arg\min_{y \in \tilde{S}} \|x - y\|^2$ satisfying

$$\langle \nabla f_\xi(x), x - x_p \rangle \geq c_{1\xi} P_\xi(x; \tilde{S}) - c_{2\xi} I_X(x),$$

where $\tilde{S} \subseteq S$, $S \subseteq \mathbb{R}^d$ is a set of global minimizers of $f$, $\tilde{S} \neq \emptyset$, $P_\xi(x; \tilde{S})$ is a nonnegative function of the argument $x \in \mathbb{R}^d$, $X \subseteq \mathbb{R}^d$, $I_X(x) := \begin{cases} 1, & \text{if } x \in X \\ 0, & \text{otherwise} \end{cases}$.

**Theorem 2.5.** *Let Assumptions 1 and 3 be satisfied. Further assume that the stepsize $\gamma^k = \min\{\tilde{\gamma}^k, \gamma_b\}$, where $\tilde{\gamma}^k$ satisfies the relations*

$$\gamma_\star \leq \tilde{\gamma}^k \leq (2 - \alpha) \frac{\langle \nabla f_{\xi^k}(x^k), x^k - x_p^k \rangle + c_{2\xi^k} I_X(x^k) + \beta_{\xi^k}^k}{\|\nabla f_{\xi^k}(x^k)\|^2}$$

*that holds for all $k \geq 0$ almost surely, where $0 < \alpha < 2$, $\beta_{\xi^k}^k \geq 0$, $\gamma_\star > 0$, $\gamma_b > 0$. Then we have the following "descent inequality" that holds for all $k \geq 0$ and the convergence result:*

$$\|x^{k+1} - x_p^{k+1}\|^2 \leq \|x^k - x_p^k\|^2 - \alpha \gamma^k c_{1\xi^k} P_{\xi^k}(x^k; \tilde{S}) + (2 - \alpha) \gamma^k \beta_{\xi^k}^k + 2\gamma^k c_{2\xi^k} I_X(x^k),$$

$$\min_{k \in \{0,...,K\}} \mathrm{E}\left[c_{1\xi^k} P_{\xi^k}(x^k; \tilde{S})\right] \leq \frac{\|x^0 - x_p^0\|^2}{\alpha \gamma_{\min}(K+1)} + C_{stoc}^K,$$

*where $C_{stoc}^K := \frac{(2-\alpha)\gamma_b}{\alpha\gamma_{\min}(K+1)} \sum_{k=0}^{K} \mathrm{E}\left[\beta_{\xi^k}^k\right] + \frac{2\gamma_b \mathrm{E}[c_{2\xi}] \sum_{k=0}^{K} \mathrm{E}[I_X(x^k)]}{\alpha\gamma_{\min}(K+1)}$, $\gamma_{\min} := \min\{\gamma_\star, \gamma_b\}$.*

**Corollary 2.6.** *Under the assumptions of Theorem 2.5 with $P(x; \tilde{S}) = f_\xi(x) - f_\xi(x_p)$, $c_{1\xi} = c_1 > 0$, if we additionally assume that $f_\xi$ are bounded from below, i.e, $f_\xi^\star := \inf_{x \in \mathbb{R}^d} f_\xi(x) > -\infty$, $f_\xi$ are L-smooth, and choose $\tilde{\gamma}^k = \frac{c_1\left(f_{\xi^k}(x^k) - f_{\xi^k}^\star\right)}{\|\nabla f(x^k)\|^2}$, $\alpha = 1$, $\beta_{\xi^k}^k = c_1(f_{\xi^k}(x_p) - f_{\xi^k}^\star)$, then we obtain*

$$\min_{k \in \{0,...,K\}} \mathrm{E}\left[f(x^k) - f^\star\right] \leq \frac{\|x^0 - x_p^0\|^2}{c_1\gamma_{\min}(K+1)} + \frac{\sigma^2\gamma_b}{\gamma_{\min}} + \frac{2\gamma_b \mathrm{E}[c_{2\xi}] \sum_{k=0}^{K} \mathrm{E}[I_X(x^k)]}{c_1\gamma_{\min}(K+1)},$$

*where $\gamma_{\min} := \min\{\frac{c_1}{2L}, \gamma_b\}$, $\sigma^2 := \mathrm{E}\left[f_\xi(x_p) - f_\xi^\star\right]$.*

**Corollary 2.7.** *Under the assumptions of Theorem 2.5, if we additionally assume that $\tilde{\gamma}^k \leq \tilde{\gamma}^{k-1}$ almost surely for $k = 1, \ldots, K$, then we obtain*

$$\min_{k \in \{0,...,K\}} \mathrm{E}\left[c_{1\xi^k} P_{\xi^k}(x^k; \tilde{S})\right] \leq \mathrm{E}\left[\frac{D_{\max}^2}{\alpha\gamma^K}\right] \frac{1}{K+1} + \tilde{C}_{stoc}^K.$$

*where $\tilde{C}_{stoc}^K := \frac{(2-\alpha)\sum_{k=0}^{K} \mathrm{E}\left[\beta_{\xi^k}^k\right] + 2\mathrm{E}[c_{2\xi}] \sum_{k=0}^{K} \mathrm{E}[I_X(x^k)]}{\alpha(K+1)}$, $D_{\max}^2 := \max_{k \in \{0,...,K\}} \|x^k - x_p^k\|^2$.*

Let us consider some examples of the choices of $P_\xi(x; \tilde{S})$, $c_{1\xi}$, $c_{2\xi}$ in Assumption 2 that describe certain classes of functions and lead to meaningful convergence results.

**Example 6.** Let $P_\xi(x; \tilde{S}) = f_\xi(x) - f_\xi(x_p)$, $c_{1\xi} = c_1 > 0$, $f_\xi^\star > -\infty$.

Note that if $\tilde{S} = S$, $S$ is a nonempty set, $c_1 = \tilde{\alpha} - \tilde{\beta}$, $c_{2\xi} = \tilde{\beta}(f_\xi(x_p) - f_\xi^\star)$, where $\tilde{\alpha} > \tilde{\beta} > 0$, then Assumption 2 is equivalent to the definition of the $\tilde{\alpha}$-$\tilde{\beta}$ condition (Islamov et al., 2024).

Let us choose $\tilde{\gamma}^k = \frac{c_1\left(f_{\xi^k}(x^k) - f_{\xi^k}^\star\right)}{\left\|\nabla f_{\xi^k}(x^k)\right\|^2}$. If we additionally assume that $f_\xi$ is $L$-smooth, then $\tilde{\gamma}^k \geq \frac{c_1}{2L}$. Thus, by setting $\alpha = 1$, $\beta_{\xi^k}^k = c_1(f_\xi(x_p) - f_\xi^\star)$, from Corollary 2.6, we get the following convergence result

$$\min_{k \in \{0,\dots,K\}} \mathrm{E}\left[f(x^k) - f^\star\right] \leq \frac{\left\|x^0 - x_p^0\right\|^2}{c_1\gamma_{\min}(K+1)} + \frac{\sigma^2\gamma_b}{\gamma_{\min}} + \frac{2\gamma_b\mathrm{E}\left[c_{2\xi}\right]\sum_{k=0}^K \mathrm{E}\left[I_X(x^k)\right]}{c_1\gamma_{\min}(K+1)}.$$

If $\sigma^2 = 0$ (in the interpolation regime) and $c_{2\xi} = 0$ (under $\tilde{\alpha}$-$\tilde{\beta}$ condition either $\tilde{\beta} = 0$, or in the interpolation regime) or $\sum_{k=0}^K \mathrm{E}\left[I_X(x^k)\right] = \mathcal{O}(1)$, then for $\min_{k \in \{0,\dots,K\}} \mathrm{E}\left[f(x^k) - f^\star\right]$ we obtain an $\mathcal{O}(1/K)$ convergence rate for under Assumption 1, 3, and the smoothness of $f_\xi$.

**Example 7.** Let $P_\xi(x; \tilde{S}) = \frac{1}{L}\|\nabla f_\xi(x)\|^2$, $L > 0$, $c_{1\xi} = c_1 > 0$.

Note that if $\tilde{S} = \{x^\star\}$, $x^\star \in S$, $c_1 = 1$, $c_{2\xi} = 0$, then Assumption 2 follows from the convexity and $L$-smoothness of functions $f_\xi$. Let us choose $\tilde{\gamma}^k = \frac{c_1 P_{\xi^k}(x^k; \tilde{S})}{\left\|\nabla f_{\xi^k}(x^k)\right\|^2} = \frac{c_1}{L}$. Then, by setting $\alpha = 1$, $\beta_{\xi^k}^k = 0$, $\tilde{\gamma}^k$ satisfies the relations of Theorem 2.5 with $\gamma_\star = \gamma_b = \frac{c_1}{L}$. Therefore,

$$\min_{k \in \{0,\dots,K\}} \mathrm{E}\left[\|\nabla f(x^k)\|^2\right] \leq \frac{L^2\left\|x^0 - x_p^0\right\|^2}{c_1^2(K+1)} + \frac{2L^2\gamma_b\mathrm{E}\left[c_{2\xi}\right]\sum_{k=0}^K \mathrm{E}\left[I_X(x^k)\right]}{c_1^2(K+1)}.$$

If $c_{2\xi} = 0$ or $\sum_{k=0}^K \mathrm{E}\left[I_X(x^k)\right] = \mathcal{O}(1)$, then we obtain an $\mathcal{O}(1/K)$ convergence rate for $\min_{k \in \{0,\dots,K\}} \mathrm{E}\left[\|\nabla f(x^k)\|^2\right]$ Assumption 1, 3, and the smoothness of $f_\xi$.

## 3 EXPERIMENTS

In the last section, we test whether our assumption holds for two choices of functions, defined by $P_\xi(x; \tilde{S}) = f_\xi(x) - f_\xi(x_p)$ and $P_\xi(x; \tilde{S}) = \|\nabla f_\xi(x)\|^2$, with $c_{1\xi} = c_1 > 0$. We assume $\tilde{S} = \{x^\star\}$, $x^K \approx x^\star \in S$. For each experiment, we use 4 different random seeds. The results demonstrate that $X$ contains only a few "problematic" points for which $c_{2\xi} > 0$, and even then $c_{2\xi}$ remains very close to zero. Consequently, by Theorem 2.5, one can obtain a reducible convergence neighborhood. All experiments were conducted in PyTorch (Paszke et al., 2019) on a single NVIDIA A100 40GB GPU.

### 3.1 HALF SPACE LEARNING PROBLEM

In the first experiment, we consider the following half-space learning problem

$$\min_{x \in \mathbb{R}^d} \left\{ f(x) := \frac{1}{n}\sum_{i=1}^n \sigma\left(-b_i x^\top a_i\right) + \frac{\lambda}{2}\|x\|^2 \right\},$$

where $\{a_i, b_i\}_{i=1}^n$, $a_i \in \mathbb{R}^d$, $b_i \in \{0, 1\}$ is a given dataset, $\lambda = 10^{-5}$, and $\sigma$ is a sigmoid function. We draw $n/2 = 20$ samples $a_i \in \mathbb{R}^8$ from two multivariate Gaussian distributions with different means and the same variance of 2, and assign the labels $b_i \in \{0, 1\}$ accordingly. We use SGD with a learning rate of 0.25 and a batch size of 1 for minimization problem.[1]

The results of the experiment are presented in Figure 1. The problem is nonconvex (Daneshmand et al., 2018), and we observe that the gradient norm becomes near zero early, indicating that the SGD trajectory passes through saddle points or local minima. From the plots, you can observe

---

[1] Our implementation is based on the open-source GitHub repository.

that for different functions $P_\xi(x; \tilde{S})$, $\mathrm{E}\left[c_{2\xi}\right]$ remains close to zero and Assumption 3 holds with relatively small constants $c_{2\xi}$ along training trajectories, when $c_1$ is fixed. Specifically, when $P_\xi(x; \tilde{S}) = f_\xi(x) - f_\xi(x_p)$, $c_1 = 1$, it follows that $c_{2\xi} \leq 0.002$, and when $P_\xi(x; \tilde{S}) = \|\nabla f_\xi(x)\|^2$, $c_1 = 0.1$, it follows that $c_{2\xi} \leq 0.058$.

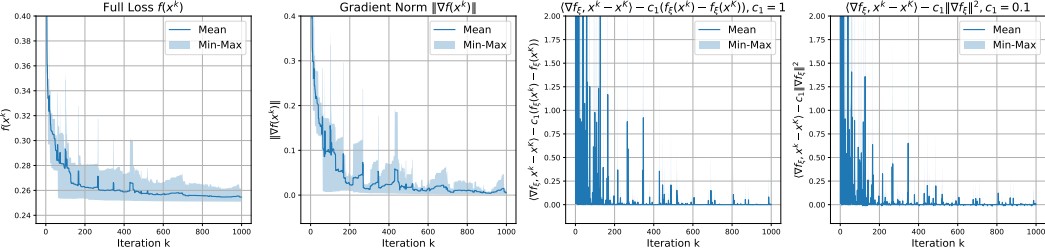

Figure 1: Training half-space learning problem.

## 3.2 MLP ARCHITECTURE

In the second experiment, we train an MLP model with 3 fully connected layers and ReLU activation functions (the second layer has a size of 64) on the Fashion-MNIST dataset (Xiao et al., 2017), using SGD with a learning rate of 0.05 and a batch size of 128. The experimental results are shown in Figure 2. The plots indicate that for different functions $P_\xi(x; \tilde{S})$, $\mathrm{E}\left[c_{2\xi}\right]$ remains close to zero and Assumption 3 holds with relatively small constants $c_{2\xi}$ along training trajectories when $c_1$ is fixed. Specifically, when $P_\xi(x; \tilde{S}) = f_\xi(x) - f_\xi(x_p)$, $c_1 = 1$, it follows that $c_{2\xi} \leq 0.402$, and when $P_\xi(x; \tilde{S}) = \|\nabla f_\xi(x)\|^2$, $c_1 = 0.1$, it follows that $c_{2\xi} \leq 0.072$.

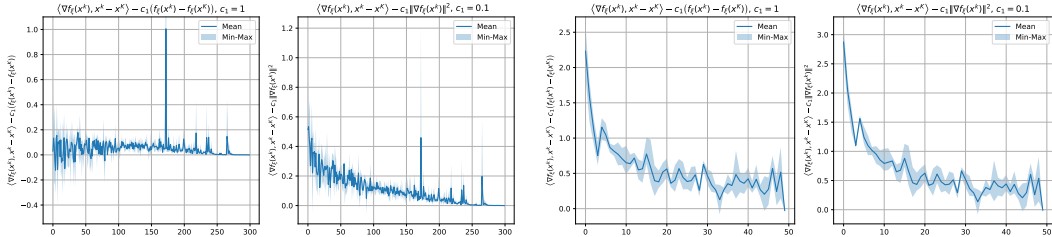

Figure 2: Training MLP model.      Figure 3: Training ResNet model.

## 3.3 RESNET ARCHITECTURE

In the final experiment, we employed the ResNet architecture (He et al., 2016) with a batch size of 128, training on the CIFAR-10 dataset (Krizhevsky, 2012) using the Adam optimizer with a learning rate of 0.001.[2] As shown in Figure 3, we observe that Assumption 3 is once again satisfied for fixed values of $c_1$, while the values of $c_{2\xi}$ remain relatively close to zero along training trajectories.

---

[2]Our implementation is based on the open-source GitHub repository.

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

Contents

## A    DETAILS AND PROOFS FROM SECTION 2.1

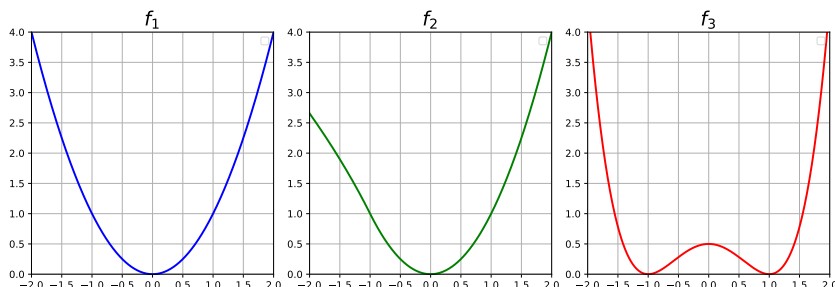

Figure 4: Examples of the function $f(x)$, $x \in \mathbb{R}$.

We consider different examples of functions $f(x)$, $x \in \mathbb{R}$ (see Figure 4):

$$f_1 = x^2, \qquad f_2 = \begin{cases} f_1, & x \geq -1 \\ 4\sqrt{-x} - 3, & x < -1 \end{cases}, \qquad f_3 = \frac{x^4}{2} - x^2 + \frac{1}{2},$$

$$f_4 = x^4 - \frac{10}{3}x^3 + 3x^2, \qquad f_5 = \begin{cases} f_4, & x \geq 0 \\ f_2, & x < 0 \end{cases}.$$

that belong to a particular class of functions. We denote each class of function in Table 2 as $F_1$, $F_2$, $F_3$, $F_4$, $F_5$, and $F_6$.

Table 3: Assumption 2 for $P(x, \tilde{S}) = f(x) - f^\star$, where $f^\star = 0$, $\tilde{S} \subseteq S$, and for different choices of $(c_1, c_2, \tilde{S})$. Here, $S \subseteq \mathbb{R}$ is a set of global minimizers of $f$, $x^\star \in S$.

| $c_1, \tilde{S}$ | $c_2 = 0$ | $c_2 \geq 0$ |
|---|---|---|
| $c_1 = 1$, $\tilde{S} = \{x^\star\}$ | $f_1 \in F_1$ | $f_1,\ f_3,\ f_4 \in F_2$ |
| $c_1 > 0$, $\tilde{S} = \{x^\star\}$ | $f_1,\ f_2,\ \in F_3$ | $f_1,\ f_2,\ f_3,\ f_4,\ f_5 \in F_4$ |
| $c_1 > 0$, $\tilde{S} = S$ | $f_1,\ f_2,\ \in F_5$ | $f_1,\ f_2,\ f_3,\ f_4,\ f_5 \in F_6$ |

1. Obviously, since $f_1$ is a convex function, we have $f_1 \in F_i$ for $i = 1, 2, 3, 4, 5, 6$.

2. The function $f_2 \in F_i$ for $i = 3, 4, 5, 6$, since
$$\langle \nabla f_2(x), x \rangle = 2\sqrt{-x} \geq c_1 f(x) = \underbrace{c_1}_{=1/2}(4\sqrt{-x} - 3), \quad \text{for} \quad x < -1.$$

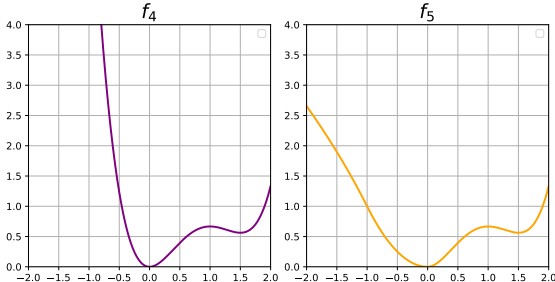

With $c_1 = 1$, it can be shown that it is not possible to satisfy this inequality by choosing any constant $c_2 \geq 0$.

3. The function $f_3 \in F_i$ for $i = 2, 4, 6$. It is easy to show $f_3$ has two global minima: a global minimum at $x = 1$ with $f^\star = 0$ and a global minimum at $x = -1$ with $f^\star = 0$. Then, choosing $c_1 = 1$ and $c_2 = 0.5$ (it is the smallest $c_2$ when $c_1 = 1$), we can show that

$$\langle \nabla f_3(x), x - 1 \rangle - c_1 f_3(x) = (2x^3 - 2x)(x - 1) - c_1 \left( \frac{x^4}{2} - x^2 + \frac{1}{2} \right) \geq -c_2, \quad \text{for} \quad x \geq 0,$$

$$\langle \nabla f_3(x), x + 1 \rangle - c_1 f_3(x) = (2x^3 - 2x)(x + 1) - c_1 \left( \frac{x^4}{2} - x^2 + \frac{1}{2} \right) \geq -c_2, \quad \text{for} \quad x < 0.$$

With $c_2 = 0$, it can be shown that it is not possible to satisfy these inequalities by choosing any constant $c_1 > 0$.

By choosing $c_1 = 1$ and $c_2 \approx 1.437$ (it is the smallest $c_2$ when $c_1 = 1$), we have

$$\langle \nabla f_3(x), x - 1 \rangle - c_1 f_3(x) = (2x^3 - 2x)(x - 1) - c_1 \left( \frac{x^4}{2} - x^2 + \frac{1}{2} \right) \geq -c_2, \quad \text{for} \quad x \in \mathbb{R}.$$

With $c_2 = 0$, it can be shown that it is not possible to satisfy this inequality by choosing any constant $c_1 > 0$.

4. The function $f_4 \in F_i$ for $i = 2, 4, 6$. It is easy to show that $f_4$ has two minima: a global minimum at $x = 0$ with $f^\star = 0$, and a local minimum at $x = 1.5$. Then, choosing $c_1 = 1$ and $c_2 \approx 1.013$ (it is the smallest $c_2$ when $c_1 = 1$), we can show that

$$\langle \nabla f_4(x), x \rangle - c_1 f_4(x) = (4 - c_1)x^4 - (10 - \frac{10}{3}c_1)x^3 + (6 - 3c_1)x^2 \geq -c_2, \quad \text{for} \quad x \in \mathbb{R}.$$

With $c_2 = 0$, it can be shown that it is not possible to satisfy this inequality by choosing any constant $c_1 > 0$.

5. The function $f_5 \in F_i$ for $i = 4, 6$. It is simply a piecewise function composed of $f_2$ and $f_4$. This statement can be easily proven using the proofs for $f_2$ and $f_4$.

# B  PROOFS FROM SECTION 2.2

## B.1  PROOF OF THEOREM 2.1

*Proof.* By the definition of $x_p$ and the gradient update, we have

$$
\begin{aligned}
\left\| x^{k+1} - x_p^{k+1} \right\|^2 &\leq \left\| x^{k+1} - x_p^k \right\|^2 \\
&= \left\| x^k - \gamma^k \nabla f\left(x^k\right) - x_p^k \right\|^2 \\
&= \left\| x^k - x_p^k \right\|^2 - 2\gamma^k \left\langle \nabla f\left(x^k\right), x^k - x_p^k \right\rangle + \left(\gamma^k\right)^2 \left\| \nabla f\left(x^k\right) \right\|^2 .
\end{aligned}
$$

Since $0 < \gamma^k \leq (2 - \alpha) \frac{\langle \nabla f(x^k), x^k - x_p^k \rangle + c_2 I_X(x^k) + \beta^k}{\| \nabla f(x^k) \|^2}$, we have

$$
\begin{aligned}
\left\| x^{k+1} - x_p^{k+1} \right\|^2 &\leq \left\| x^k - x_p^k \right\|^2 - \alpha \gamma^k \left\langle \nabla f\left(x^k\right), x^k - x_p^k \right\rangle + (2 - \alpha)\beta^k \gamma^k \\
&\quad + (2 - \alpha)c_2 I_X(x^k)\gamma^k \\
&\overset{2}{\leq} \left\| x^k - x_p^k \right\|^2 - \alpha c_1 \gamma^k P(x^k; \tilde{S}) + (2 - \alpha)\beta^k \gamma^k + 2c_2 I_X(x^k)\gamma^k .
\end{aligned}
$$

After telescoping the last inequality, we get

$$
\begin{aligned}
\min_{k \in \{0, \dots, K\}} P(x^k; \tilde{S}) &\leq \frac{\sum_{k=0}^K \gamma^k P(x^k; \tilde{S})}{\sum_{k=0}^K \gamma^k} \\
&\leq \frac{\left\| x^0 - x_p^0 \right\|^2}{\alpha c_1 \sum_{k=0}^K \gamma^k} + \frac{\sum_{k=0}^K \gamma^k (2 - \alpha)\beta^k}{\alpha c_1 \sum_{k=0}^K \gamma^k} + \frac{2c_2 \sum_{k=0}^K \gamma^k I_X(x^k)}{\alpha c_1 \sum_{k=0}^K \gamma^k} .
\end{aligned}
$$

$\square$

## B.2 PROOF OF COROLLARY 2.2

*Proof.* If $P(x; \tilde{S}) = f(x) - f^\star$ and $\gamma^k = \frac{c_1 P(x^k; \tilde{S})}{\|\nabla f(x^k)\|^2} = \frac{c_1 (f(x^k) - f^\star)}{\|\nabla f(x^k)\|^2}$, then, by setting $\alpha = 1$, $\beta^k = 0$, $\gamma^k$ satisfies the relations of Theorem 2.1

$$0 < \gamma^k \overset{2}{\leq} \frac{\langle \nabla f(x^k), x^k - x^\star \rangle + c_2 I_X(x^k)}{\|\nabla f(x^k)\|^2}.$$

If we additionally assume that $f$ is $L$-smooth function, we can show

$$\gamma^k = \frac{c_1 (f(x^k) - f^\star)}{\|\nabla f(x^k)\|^2} \geq \frac{c_1 \frac{1}{2L} \|\nabla f(x^k)\|^2}{\|\nabla f(x^k)\|^2} \geq \frac{c_1}{2L}.$$

Indeed, consider function $\varphi(y) = f(y) - f^\star$, then $\varphi(y) \geq 0$ for all $y \in \mathbb{R}^d$. Using smoothness of $\varphi$ and choosing $y = x - \frac{1}{L}\nabla f(x)$, we get

$$0 \leq \varphi(y) \leq \varphi(x) + \langle \nabla \varphi(x), y - x \rangle + \frac{L}{2}\|y - x\|^2 = f(x) - f^\star - \frac{1}{2L}\|\nabla f(x)\|^2.$$

Therefore, from Theorem 2.1 we get the following convergence result

$$\min_{k \in \{0, \dots, K\}} f(x^k) - f^\star \leq \frac{2L \|x^0 - x_p^0\|^2}{c_1^2(K+1)} + \frac{4c_2 L \sum_{k=0}^{K} \gamma^k I_X(x^k)}{c_1^2(K+1)}.$$

$\square$

## B.3 PROOF OF COROLLARY 2.3

*Proof.* Let us choose $\gamma^k = (2 - \alpha)\frac{c_1 P(x^k; \tilde{S}) + \beta^k}{\|\nabla f(x^k)\|^2}$, then

$$0 < \gamma^k \overset{2}{\leq} (2 - \alpha)\frac{\langle \nabla f(x^k), x^k - x_p^k \rangle + c_2 I_X(x^k) + \beta^k}{\|\nabla f(x^k)\|^2}.$$

From Theorem 2.1 we have

$$\frac{\sum_{k=0}^{K} \gamma^k P(x^k; \tilde{S})}{\sum_{k=0}^{K} \gamma^k} \leq \frac{\|x^0 - x_p^0\|^2}{\alpha c_1 \sum_{k=0}^{K} \gamma^k} + C^K,$$

or equivalently

$$\sum_{k=0}^{K} \gamma^k \left( P(x^k; \tilde{S}) - C^K \right) \leq \frac{\|x^0 - x_p^0\|^2}{\alpha c_1}.$$

Since $\gamma^k \geq (2 - \alpha)\frac{c_1 P(x^k; \tilde{S})}{\|\nabla f(x^k)\|^2}$, we have

$$\sum_{k=0}^{K} \frac{P(x^k; \tilde{S}) \left( P(x^k; \tilde{S}) - C^K \right)}{\|\nabla f(x^k)\|^2} \leq \frac{\|x^0 - x_p^0\|^2}{(2 - \alpha)\alpha c_1^2}.$$

Let us assume that $P(x^k; \tilde{S}) > C^K$ for $k = 0, \dots, K$, otherwise, $\min_{k \in \{0, \dots, K\}} P(x^k; \tilde{S}) \leq C^K$. If we also assume that $f$ has bounded gradients, i.e., $\|\nabla f(x)\| \leq G$ for all $x \in \mathbb{R}^d$, then we get

$$\sum_{k=0}^{K} \frac{\left( P(x^k; \tilde{S}) - C^K \right)^2}{G^2} \leq \sum_{k=0}^{K} \frac{P(x^k; \tilde{S}) \left( P(x^k; \tilde{S}) - C^K \right)}{\|\nabla f(x^k)\|^2} \leq \frac{\|x^0 - x_p^0\|^2}{(2 - \alpha)\alpha c_1^2},$$

consequently,

$$\min_{k \in \{0, \dots, K\}} \left( P(x^k; \tilde{S}) - C^K \right)^2 \leq \frac{G^2 \|x^0 - x_p^0\|^2}{(2 - \alpha)\alpha c_1^2} \frac{1}{K+1},$$

or equivalently

$$\min_{k \in \{0, \dots, K\}} P(x^k; \tilde{S}) \leq \frac{G \|x^0 - x_p^0\|}{\sqrt{(2 - \alpha)\alpha} c_1} \frac{1}{\sqrt{K+1}} + C^K.$$

$\square$

### B.4 PROOF OF COROLLARY 2.4

*Proof.* From Theorem 2.1 we have the following descent inequality

$$\left\| x^{k+1} - x_p^{k+1} \right\|^2 \leq \left\| x^k - x_p^k \right\|^2 - \alpha c_1 \gamma^k P(x^k; \tilde{S}) + (2 - \alpha)\beta^k \gamma^k + 2c_2 I_X(x^k)\gamma^k.$$

If $\gamma^k \leq \gamma^{k-1}$, instead of immediate telescoping the descent inequality, we can divide it by $\alpha c_1 \gamma^k$ and then telescope

$$\sum_{k=0}^{K} P(x^k; \tilde{S}) \leq \sum_{k=0}^{K} \frac{\left\| x^k - x_p^k \right\|^2}{\alpha c_1 \gamma^k} - \sum_{k=0}^{K} \frac{\left\| x^{k+1} - x_p^{k+1} \right\|^2}{\alpha c_1 \gamma^k} + \sum_{k=0}^{K} \frac{(2 - \alpha)\beta^k + 2c_2 I_X(x^k)}{\alpha c_1}$$

$$\leq \frac{\left\| x^0 - x_p^0 \right\|^2}{\alpha c_1 \gamma^0} + \sum_{k=1}^{K} \frac{\left\| x^k - x_p^k \right\|^2}{\alpha c_1 \gamma^k} - \sum_{k=1}^{K} \frac{\left\| x^k - x_p^k \right\|^2}{\alpha c_1 \gamma^{k-1}}$$

$$+ \sum_{k=0}^{K} \frac{(2 - \alpha)\beta^k + 2c_2 I_X(x^k)}{\alpha c_1}$$

$$\overset{\gamma^k \leq \gamma^{k-1}}{\leq} \frac{D_{\max}^2}{\alpha c_1} \left( \frac{1}{\gamma^0} + \sum_{k=1}^{K} \left( \frac{1}{\gamma^k} - \frac{1}{\gamma^{k-1}} \right) \right) + \sum_{k=0}^{K} \frac{(2 - \alpha)\beta^k + 2c_2 I_X(x^k)}{\alpha c_1}$$

$$= \frac{D_{\max}^2}{\alpha c_1 \gamma^K} + \sum_{k=0}^{K} \frac{(2 - \alpha)\beta^k + 2c_2 I_X(x^k)}{\alpha c_1},$$

where $D_{\max}^2 := \max_{k \in \{0, \dots, K\}} \left\| x^k - x_p^k \right\|^2$.

Therefore, we obtain

$$\min_{k \in \{0, \dots, K\}} P(x^k; \tilde{S}) \leq \frac{D_{\max}^2}{\alpha c_1 \gamma^K (K + 1)} + \frac{2 - \alpha}{\alpha c_1 (K + 1)} \sum_{k=0}^{K} \beta^k + \frac{2c_2}{\alpha c_1 (K + 1)} \sum_{k=0}^{K} I_X(x^k).$$

$\square$

## C PROOFS FOR EXAMPLES OF STEPSIZES FROM SECTION 2.3

**Case 1.** Let us choose $\gamma^k = \frac{c_1 (f(x^k) - f^\star)}{\|\nabla f(x^k)\|^2}$. Then, by setting $\alpha = 1$, $\beta^k = 0$, $\gamma^k$ satisfies the relations of Theorem 2.1. If we assume that $f$ is $L$-smooth function, we can show

$$\gamma^k = \frac{c_1 \left( f(x^k) - f^\star \right)}{\|\nabla f(x^k)\|^2} \geq \frac{c_1 \frac{1}{2L} \|\nabla f(x^k)\|^2}{\|\nabla f(x^k)\|^2} \geq \frac{c_1}{2L}.$$

Indeed, consider function $\varphi(y) = f(y) - f^\star$, then $\nabla \varphi(x^\star) = 0$ and $\varphi(y) \geq \varphi(x^\star) = 0$ for all $y \in \mathbb{R}^d$. Using smoothness of $\varphi$ and choosing $y = x - \frac{1}{L}\nabla f(x)$, we get

$$0 \leq \varphi(y) \leq \varphi(x) + \langle \nabla \varphi(x), y - x \rangle + \frac{L}{2} \|y - x\|^2$$

$$= f(x) - f^\star - \frac{1}{2L} \|\nabla f(x)\|^2.$$

Therefore, the convergence rate will be

$$\min_{k \in \{0, \dots, K\}} f(x^k) - f^\star \leq \frac{2L \left\| x^0 - x^\star \right\|^2}{c_1^2 (K + 1)} + \frac{2c_2 \sum_{k=0}^{K} \gamma^k I_X(x^k)}{c_1 \sum_{k=0}^{K} \gamma^k}.$$

**Case 2.** Let us choose $\gamma^k = \frac{c_1 (f(x^k) - l^\star)}{\|\nabla f(x^k)\|^2}$, $l^\star \leq f^\star$. Then, by setting $\alpha = 1$, $\beta^k = c_1 (f^\star - l^\star)$, $\gamma^k$ satisfies the relations of Theorem 2.1. If we assume that $f$ is $L$-smooth function, we can show

$$\gamma^k = \frac{c_1 \left( f(x^k) - l^\star \right)}{\|\nabla f(x^k)\|^2} \geq \frac{c_1 \left( f(x^k) - f^\star \right)}{\|\nabla f(x^k)\|^2} \geq \frac{c_1}{2L}.$$

Therefore, the convergence rate will be

$$\min_{k\in\{0,...,K\}} f(x^k) - f^\star \leq \frac{2L\left\|x^0 - x^\star\right\|^2}{c_1^2(K+1)} + \sigma^2 + \frac{2c_2\sum_{k=0}^{K}\gamma^k I_X(x^k)}{c_1\sum_{k=0}^{K}\gamma^k},$$

where $\sigma^2 := f^\star - l^\star$.

**Case 3.** Let us choose

$$\gamma^k = \frac{1}{c^k}\min\left\{\frac{c_1\left(f(x^k) - l^\star\right)}{\left\|\nabla f\left(x^k\right)\right\|^2}, \gamma^{k-1}c^{k-1}\right\},$$

where $l^\star \leq f^\star$, $\{c^k\}$ is any non-decreasing sequence such that $c^k \geq 1$, $c^{-1} = c^0$, $\gamma^{-1} = \gamma^0 > 0$. First, note that $\gamma^k \leq \gamma^{k-1}$ holds. Then, by setting $\alpha = 1$, $\beta^k = \frac{c_1}{c^k}(f^\star - l^\star)$, $\gamma^k$ satisfies the relations of Theorem 2.1

$$0 < \gamma^k \leq \frac{1}{c^k}\frac{c_1\left(f(x^k) - l^\star\right)}{\left\|\nabla f\left(x^k\right)\right\|^2} \overset{2}{\leq} \frac{\left\langle\nabla f\left(x^k\right), x^k - x^\star\right\rangle + c_2 I_X(x^k) + \beta^k}{\left\|\nabla f\left(x^k\right)\right\|^2}.$$

If we assume that $f$ is $L$-smooth function, we can show recursively that

$$\gamma^K = \frac{1}{c^K}\min\left\{\frac{c_1\left(f(x^K) - l^\star\right)}{\left\|\nabla f\left(x^K\right)\right\|^2}, \gamma^{K-1}c^{K-1}\right\}$$

$$\geq \min\left\{\frac{c_1\left(f(x^K) - f^\star\right)}{c^K\left\|\nabla f\left(x^K\right)\right\|^2}, \frac{\gamma^{K-1}c^{K-1}}{c^K}\right\}$$

$$\geq \min\left\{\frac{c_1}{2c^K L}, \frac{\gamma^{K-1}c^{K-1}}{c^K}\right\}$$

$$\geq \min\left\{\frac{c_1}{2c^K L}, \ldots, \frac{c_1}{2c^0 L}, \frac{\gamma^0 c^0}{c^K}\right\}$$

$$\ldots$$

$$\geq \min\left\{\frac{c_1}{2c^K L}, \frac{\gamma^0 c^0}{c^K}\right\} = \frac{c_1}{2c^K \tilde{L}},$$

where $\tilde{L} = \max\left\{L, \frac{c_1}{2\gamma^0 c^0}\right\}$.

Therefore, the convergence rate will be

$$\min_{k\in\{0,...,K\}} f(x^k) - f^\star \leq \frac{D^2}{c_1\gamma^K(K+1)} + \frac{1}{c_1(K+1)}\underbrace{\sum_{k=0}^{K}\beta^k}_{=\sum_{k=0}^{K}\frac{c_1\sigma^2}{c^k}} + \frac{2c_2\sum_{k=0}^{K}\gamma^k I_X(x^k)}{c_1\sum_{k=0}^{K}\gamma^k}$$

$$\leq \frac{2D^2 c^K \tilde{L}}{c_1^2(K+1)} + \frac{\sigma^2}{K+1}\sum_{k=0}^{K}\frac{1}{c^k} + \frac{2c_2\sum_{k=0}^{K}\gamma^k I_X(x^k)}{c_1\sum_{k=0}^{K}\gamma^k}$$

$$\overset{c^k=\sqrt{1+k}}{\leq} \frac{2D^2 c^K \tilde{L}}{c_1^2(K+1)} + \frac{2\sigma^2}{\sqrt{K+1}} + \frac{2c_2\sum_{k=0}^{K}\gamma^k I_X(x^k)}{c_1\sum_{k=0}^{K}\gamma^k}.$$

where $\sigma^2 = f^\star - l^\star$.

## D  ADDITIONAL EXAMPLES OF FUNCTION CLASSES FROM SECTION 2.4

***Example 4.*** Let $P(x;\tilde{S}) = f(x) - f^\star + \frac{\mu}{2}\|x - x_p\|^2$, $\mu > 0$.

Note that if $\tilde{S} = \{x^\star\}$, $x^\star \in S$, $c_2 = 0$, then Assumption 2 is equivalent to the definition of $\mu$-strongly $c_1$-weak quasi-convex functions (Bu & Mesbahi, 2020). If additionally $c_1 = 1$, then Assumption 2 follows from the $\mu$-strong convexity of the function $f$.

Let us choose $\gamma^k = \frac{c_1\left(f(x^k)-f^\star\right)}{\|\nabla f(x^k)\|^2} \leq \frac{c_1 P(x^k;\tilde{S})}{\|\nabla f(x^k)\|^2}$. Then, by setting $\alpha = 1$, $\beta^k = 0$, $\gamma^k$ satisfies the relations of Theorem 2.1

$$0 < \gamma^k \leq \frac{c_1 P(x^k;\tilde{S})}{\|\nabla f(x^k)\|^2} \overset{2}{\leq} \frac{\langle \nabla f(x^k), x^k - x_p^k \rangle + c_2 I_X(x^k)}{\|\nabla f(x^k)\|^2}.$$

Similar to the previous example, assuming that $f$ is $L$-smooth, we can show that

$$\min_{k\in\{0,...,K\}} f(x^k) - f^\star + \frac{\mu}{2}\|x^k - x_p^k\|^2 \leq \frac{2L\|x^0 - x_p^k\|^2}{c_1^2(K+1)} + \frac{2c_2\sum_{k=0}^K \gamma^k I_X(x^k)}{c_1\sum_{k=0}^K \gamma^k},$$

and assuming that $f$ has bounded gradients, we get

$$\min_{k\in\{0,...,K\}} f(x^k) - f^\star + \frac{\mu}{2}\|x^k - x_p^k\|^2 \leq \frac{G\|x^0 - x_p^k\|}{c_1\sqrt{K+1}} + \frac{2c_2\sum_{k=0}^K \gamma^k I_X(x^k)}{c_1\sum_{k=0}^K \gamma^k}.$$

If $c_2 = 0$ or $\sum_{k=0}^K \gamma^k I_X(x^k) = \mathcal{O}(1)$, then we obtain an $\mathcal{O}\left(\frac{1}{K}\right)$ convergence rate for $\min_{k\in\{0,...,K\}} f(x^k) - f^\star + \frac{\mu}{2}\|x^k - x_p^k\|^2$ under Assumptions 1, 2, and the smoothness of $f$, and an $\mathcal{O}\left(\frac{1}{\sqrt{K}}\right)$ convergence rate for $\min_{k\in\{0,...,K\}} f(x^k) - f^\star + \frac{\mu}{2}\|x^k - x_p^k\|^2$ under Assumptions 1, 2, and the boundedness of the gradients of $f$.

We can also establish the linear rate of convergence for this class of functions. Indeed, using the descent inequality from Theorem 2.1 and assuming $f$ is $L$-smooth, we get

$$\|x^{k+1} - x_p^{k+1}\|^2 \leq \|x^k - x_p^k\|^2 - c_1\gamma^k P(x^k;\tilde{S}) + 2c_2 I_X(x^k)\gamma^k$$

$$\leq \left(1 - \frac{c_1\gamma^k\mu}{2}\right)\|x^k - x_p^k\|^2 + 2c_2 I_X(x^k)\gamma^k$$

$$\leq \left(1 - \frac{c_1^2\mu}{4L}\right)\|x^k - x_p^k\|^2 + 2c_2 I_X(x^k)\gamma^k.$$

Therefore, we finally get the following convergence result

$$\|x^K - x_p^K\|^2 \leq \left(1 - \frac{c_1^2\mu}{4L}\right)^K \|x^0 - x_p^0\|^2 + \frac{8c_2 L\gamma_{\max}}{c_1^2\mu},$$

where $\gamma_{\max} := \max_{k\in 0,...,K} \gamma^k I_X(x^k)$.

If $c_2 = 0$, then we obtain a linear convergence rate for $\|x^K - x_p^K\|^2$ under Assumptions 1, 2, and the smoothness of $f$.

***Example 5.*** Let $P(x;\tilde{S}) = f(x) - f^\star + \frac{1}{2L}\|\nabla f(x)\|^2$, $L > 0$.

Note that if $\tilde{S} = \{x^\star\}$, $x^\star \in S$, $c_1 = 1$, $c_2 = 0$, then Assumption 2 follows from the convexity and $L$-smoothness of the function $f$. Here, we used the fact that $f$ is $L$-smooth and convex , which is equivalent to the property that for all $x, y \in \mathbb{R}^d$

$$\frac{1}{2L}\|\nabla f(x) - \nabla f(y)\|^2 \leq f(x) - f(y) - \langle \nabla f(y), x - y \rangle.$$

Let us choose $\gamma^k = \frac{c_1 P(x^k;\tilde{S})}{\|\nabla f(x^k)\|^2} = \frac{c_1\left(f(x^k)-f^\star\right)}{\|\nabla f(x^k)\|^2} + \frac{c_1}{2L}$. Then, by setting $\alpha = 1$, $\beta^k = 0$, $\gamma^k$ satisfies the relations of Theorem 2.1

$$0 < \gamma^k \overset{2}{\leq} \frac{\langle \nabla f(x^k), x^k - x_p^k \rangle + c_2 I_X(x^k)}{\|\nabla f(x^k)\|^2}.$$

Therefore, using the fact that $\sum_{k=0}^K \gamma^k \geq \frac{c_1}{2L}(K+1)$, from Theorem 2.1 we get the following convergence result

$$\min_{k\in\{0,...,K\}} f(x^k) - f^\star + \frac{1}{2L}\|\nabla f(x^k)\|^2 \leq \frac{2L\|x^0 - x_p^0\|^2}{c_1^2(K+1)} + \frac{2c_2\sum_{k=0}^K \gamma^k I_X(x^k)}{c_1\sum_{k=0}^K \gamma^k}.$$

If $c_2 = 0$ or $\sum_{k=0}^K \gamma^k I_X(x^k) = \mathcal{O}(1)$, then we obtain an $\mathcal{O}\left(\frac{1}{K}\right)$ convergence rate for $\min_{k\in\{0,...,K\}} f(x^k) - f^\star + \frac{1}{2L}\|\nabla f(x^k)\|^2$ under Assumptions 1, 2.

# E PROOFS FROM SECTION 2.5

## E.1 PROOF OF THEOREM 2.5

*Proof.* By the definition of $x_p$ and the gradient update, we have

$$\left\|x^{k+1} - x_p^{k+1}\right\|^2 \leq \left\|x^{k+1} - x_p^k\right\|^2$$
$$= \left\|x^k - \gamma^k \nabla f_{\xi^k}\left(x^k\right) - x_p^k\right\|^2$$
$$= \left\|x^k - x_p^k\right\|^2 - 2\gamma^k \left\langle \nabla f_{\xi^k}\left(x^k\right), x^k - x_p^k\right\rangle + \left(\gamma^k\right)^2 \left\|\nabla f_{\xi^k}\left(x^k\right)\right\|^2.$$

Since $\gamma_\star \leq \tilde{\gamma}^k \leq (2-\alpha)\frac{\langle \nabla f_{\xi^k}(x^k), x^k - x_p^k\rangle + c_2{}_{\xi^k} I_X(x^k) + \beta_{\xi^k}^k}{\left\|\nabla f_{\xi^k}(x^k)\right\|^2}$ and $\gamma^k = \min\{\tilde{\gamma}^k, \gamma_b\}$, we have

$$\left\|x^{k+1} - x_p^{k+1}\right\|^2 \leq \left\|x^k - x_p^k\right\|^2 - 2\gamma^k \left\langle \nabla f_{\xi^k}\left(x^k\right), x^k - x_p^k\right\rangle \mid x^k + \gamma^k \tilde{\gamma}^k \left\|\nabla f_{\xi^k}\left(x^k\right)\right\|^2$$
$$\leq \left\|x^k - x_p^k\right\|^2 - 2\gamma^k \left\langle \nabla f_{\xi^k}\left(x^k\right), x^k - x_p^k\right\rangle$$
$$+ (2-\alpha)\gamma^k \left\langle \nabla f_{\xi^k}\left(x^k\right), x^k - x_p^k\right\rangle + (2-\alpha)\left(\gamma^k \beta_{\xi^k}^k + \gamma^k c_2{}_{\xi^k} I_X(x^k)\right)$$
$$= \left\|x^k - x_p^k\right\|^2 - \alpha\gamma^k \left\langle \nabla f_{\xi^k}\left(x^k\right), x^k - x_p^k\right\rangle$$
$$+ (2-\alpha)\left(\gamma^k \beta_{\xi^k}^k + \gamma^k c_2{}_{\xi^k} I_X(x^k)\right)$$
$$\overset{3}{\leq} \left\|x^k - x_p^k\right\|^2 - \alpha\gamma^k c_1{}_{\xi^k} P_{\xi^k}(x^k; \tilde{S}) + \alpha\gamma^k c_2{}_{\xi^k} I_X(x^k)$$
$$+ (2-\alpha)\left(\gamma^k \beta_{\xi^k}^k + \gamma^k c_2{}_{\xi^k} I_X(x^k)\right)$$
$$= \left\|x^k - x_p^k\right\|^2 - \alpha\gamma^k c_1{}_{\xi^k} P_{\xi^k}(x^k; \tilde{S}) + (2-\alpha)\gamma^k \beta_{\xi^k}^k + 2\gamma^k c_2{}_{\xi^k} I_X(x^k)$$
$$\leq \left\|x^k - x_p^k\right\|^2 - \alpha\gamma_{\min} c_1{}_{\xi^k} P_{\xi^k}(x^k; \tilde{S}) + (2-\alpha)\gamma_b \beta_{\xi^k}^k + 2\gamma_b c_2{}_{\xi^k} I_X(x^k),$$

where $\gamma_{\min} := \min\{\gamma_\star, \gamma_b\}$.

By taking expectation, we have

$$\mathrm{E}\left[\left\|x^{k+1} - x_p^{k+1}\right\|^2\right] \leq \mathrm{E}\left[\left\|x^k - x_p^k\right\|^2\right] - \alpha\gamma_{\min}\mathrm{E}\left[c_1{}_{\xi^k} P_{\xi^k}(x^k; \tilde{S})\right]$$
$$+ (2-\alpha)\gamma_b \mathrm{E}\left[\beta_{\xi^k}^k\right] + 2\gamma_b \mathrm{E}\left[c_2{}_\xi\right]\mathrm{E}\left[I_X(x^k)\right].$$

After telescoping the last inequality, we get

$$\min_{k\in\{0,\ldots,K\}} \mathrm{E}\left[c_1{}_{\xi^k} P_{\xi^k}(x^k; \tilde{S})\right] \leq \frac{\left\|x^0 - x_p^0\right\|^2}{\alpha\gamma_{\min}(K+1)} + \frac{(2-\alpha)\gamma_b}{\alpha\gamma_{\min}(K+1)}\sum_{k=0}^K \mathrm{E}\left[\beta_{\xi^k}^k\right]$$
$$+ \frac{2\gamma_b \mathrm{E}\left[c_2{}_\xi\right]\sum_{k=0}^K \mathrm{E}\left[I_X(x^k)\right]}{\alpha\gamma_{\min}(K+1)}.$$

□

## E.2 PROOF OF COROLLARY 2.6

*Proof.* It follows straightforwardly from Theorem 2.5 and using the smoothness of $f_\xi$, since

$$\tilde{\gamma}^k = \frac{c_1\left(f_{\xi^k}(x^k) - f_{\xi^k}^\star\right)}{\left\|\nabla f_{\xi^k}\left(x^k\right)\right\|^2} \geq \frac{c_1}{2L} = \gamma_\star.$$

□

### E.3 Proof of Corollary 2.7

*Proof.* From Theorem 2.5 we have the following descent inequality

$$\left\|x^{k+1} - x_p^{k+1}\right\|^2 \le \left\|x^k - x_p^k\right\|^2 - \alpha\gamma^k c_{1\xi^k} P_{\xi^k}(x^k; \tilde{S}) + (2-\alpha)\gamma^k \beta_{\xi^k}^k + 2\gamma^k c_{2\xi^k} I_X(x^k).$$

If $\tilde{\gamma}^k \le \tilde{\gamma}^{k-1}$, then $\gamma^k \le \gamma^{k-1}$ and instead of immediate telescoping the descent inequality, we can divide it by $\alpha\gamma^k$ and then telescope

$$
\begin{aligned}
\sum_{k=0}^K c_{1\xi^k} P_{\xi^k}(x^k; \tilde{S}) &\le \sum_{k=0}^K \frac{\left\|x^k - x_p^k\right\|^2}{\alpha\gamma^k} - \sum_{k=0}^K \frac{\left\|x^{k+1} - x_p^{k+1}\right\|^2}{\alpha\gamma^k} \\
&\quad + \sum_{k=0}^K \frac{(2-\alpha)\beta_{\xi^k}^k + 2c_{2\xi^k} I_X(x^k)}{\alpha} \\
&\le \frac{\left\|x^0 - x_p^0\right\|^2}{\alpha\gamma^0} + \sum_{k=1}^K \frac{\left\|x^k - x_p^k\right\|^2}{\alpha\gamma^k} - \sum_{k=1}^K \frac{\left\|x^k - x_p^k\right\|^2}{\alpha\gamma^{k-1}} \\
&\quad + \sum_{k=0}^K \frac{(2-\alpha)\beta_{\xi^k}^k + 2c_{2\xi^k} I_X(x^k)}{\alpha} \\
&\stackrel{\gamma^k \le \gamma^{k-1}}{\le} \frac{D_{\max}^2}{\alpha}\left(\frac{1}{\gamma^0} + \sum_{k=1}^K \left(\frac{1}{\gamma^k} - \frac{1}{\gamma^{k-1}}\right)\right) \\
&\quad + \sum_{k=0}^K \frac{(2-\alpha)\beta_{\xi^k}^k + 2c_{2\xi^k} I_X(x^k)}{\alpha} \\
&= \frac{D_{\max}^2}{\alpha\gamma^K} + \sum_{k=0}^K \frac{(2-\alpha)\beta_{\xi^k}^k + 2c_{2\xi^k} I_X(x^k)}{\alpha},
\end{aligned}
$$

where $D_{\max}^2 := \max_{k\in\{0,\dots,K\}} \left\|x^k - x_p^k\right\|^2$.

After taking expectation, we have

$$\sum_{k=0}^K \mathrm{E}\left[c_{1\xi^k} P_{\xi^k}(x^k; \tilde{S})\right] \le \mathrm{E}\left[\frac{D_{\max}^2}{\alpha\gamma^K}\right] + \sum_{k=0}^K \frac{(2-\alpha)\mathrm{E}\left[\beta_{\xi^k}^k\right] + 2\mathrm{E}\left[c_{2\xi}\right]\mathrm{E}\left[I_X(x^k)\right]}{\alpha},$$

Therefore, we obtain

$$
\begin{aligned}
\min_{k\in\{0,\dots,K\}} \mathrm{E}\left[c_{1\xi^k} P_{\xi^k}(x^k; \tilde{S})\right] &\le \mathrm{E}\left[\frac{D_{\max}^2}{\alpha\gamma^K}\right]\frac{1}{K+1} + \frac{2-\alpha}{\alpha(K+1)}\sum_{k=0}^K \mathrm{E}\left[\beta_{\xi^k}^k\right] \\
&\quad + \frac{2\mathrm{E}\left[c_{2\xi}\right]\sum_{k=0}^K \mathrm{E}\left[I_X(x^k)\right]}{\alpha(K+1)}.
\end{aligned}
$$

$\square$

