# OpenReview forum: "A Novel Unified Parametric Assumption for Nonconvex Optimization"
_ICLR.cc/2026/Conference — Submitted to ICLR 2026_

### Official Review · Reviewer_hsei · 2025-10-19

**Soundness:** 2
**Presentation:** 2
**Contribution:** 1
**Rating:** 2
**Confidence:** 4

**Summary:**

This paper provides a unified parametric assumption to study the convergence of several problems, including PL and weakly quasi-convex problems. They derive convergence theorems for both deterministic and stochastic optimization, and conduct numerical experiments.

**Strengths:**

1. They provide a novel parametric assumption that encompasses several nonconvex problem classes.

2. They provide both theoretical justification and empirical evidence.

**Weaknesses:**

1. I find the significance of Assumption 2 unclear. It appears to cover only a limited class of nonconvex optimization problems, such as those satisfying the PL condition or quasi-convexity. However, most modern deep learning applications—such as CNNs and LLM training—are substantially more complex and fall outside the scope of this assumption. Moreover, the goals stated in the introduction and abstract seem overly ambitious: they claim that existing classical research can only guarantee convergence to an $\epsilon$-stationary point or a higher-order stationary point, as finding a global minimum of a general nonconvex function is NP-hard [1]. Yet, the present work does not address these fundamental challenges. The restrictiveness of Assumption 2 limits the generality of the proposed method and offers little new insight into modern deep learning problems.

**References**

- [1] Murty, K. G., & Kabadi, S. N. (1985). Some NP-complete problems in quadratic and nonlinear programming.

**Questions:**

See the weakness.

---

### Official Review · Reviewer_jtzj · 2025-10-28

**Soundness:** 3
**Presentation:** 2
**Contribution:** 3
**Rating:** 4
**Confidence:** 4

**Summary:**

This paper introduces a unified parametric assumption for nonconvex optimization that aims to bridge the gap between general nonconvex functions and more structured function classes. The authors provide convergence guarantees for gradient-based methods under this assumption and demonstrate its applicability with examples. The paper is generally well-written and provides an interesting theoretical perspective on unifying several existing assumptions in nonconvex optimization.

**Strengths:**

* They introduce a novel unified parametric assumption that can encompass a wide range of nonconvex functions, offering a fresh theoretical perspective.

* They provide convergence guarantees for both deterministic and stochastic gradient-based methods.

* The examples and discussion help illustrate the practical relevance of the proposed assumption.

**Weaknesses:**

* Could the authors clarify the meaning of the set $\mathcal{X}$ in Assumption 2? Is there a relationship between $\mathcal{X}$ and $\mathcal{S}$? It would be helpful to provide intuition for why $\mathcal{X}$ is introduced and how it should be chosen in practice.

* The progress function $P(x;\tilde{\mathcal{S}})$ (even the set $\mathcal{X}$) depends on $\mathcal{S}$. However, for functions satisfying Assumption 1, finding the global minimizer is NP-hard. How can Assumption 2 be verified in practice? This suggests that the assumption may be somewhat restrictive.

* Many practical problems involve constraints. Could Assumption 2 be extended to constrained settings?

* Could the authors provide an example satisfying the PL condition? If so, the assumption may also encompass the KL condition for differentiable functions.

* In the convergence analysis, additional parameters $\alpha$ and $\beta_k$ are introduced. In the deterministic setting, choosing $\alpha=1$ and $\beta_k=0$ suffices for convergence, and in the stochastic setting, $\alpha=1$ may still be sufficient. To make the theorems more elegant and informative, could these parameters be simplified or removed?

**Questions:**

Please see above.

---

### Official Review · Reviewer_74WJ · 2025-10-30

**Soundness:** 2
**Presentation:** 2
**Contribution:** 2
**Rating:** 4
**Confidence:** 4

**Summary:**

The paper introduces a new parametric assumption that encompasses a wide class of non-convex functions and unifies several existing assumptions in the literature. It establishes a general convergence result under this framework and recovers several known results. The paper also provides empirical measurements of the assumption in neural network training settings.

**Strengths:**

- Explores a direction of closing the gap between empirical and theoretical optimization in NN training.
- The idea of allowing outlier points in the assumption and getting bound based on the occurrence of those is interesting.

**Weaknesses:**

- The presentation and flaw of the paper can be improved.
- I am not sure the theoretical contribution is novel. The main theorem is a direct application of descent lemma.
- The subsequent corollaries give existing standard results including a penalty term of outliers.
- I am not convinced by the experimental results.

**Questions:**

- Line 031. Why do you mention scaling law? The success of first order optimization in training NNs has been around much before scaling laws. How is it related here?
- After assumption 2, it would be good to discuss the role of the set X, saying it represents outlier points or something like that.
- You mention smoothness several times in the manuscript. Would be good to define it.
- Section 1.1. You motivate why you mention about alternatives to convexity but I cannot see why do you mention alternatives to smoothness. The flaw of text there is unnatural.
- Typo Line 120. You need to say $x \in X$ for your condition to hold. And thus that holds for an outlier point which doesn't say much.
- Can you clarify why do you consider the functions in the "Role of parameters section". The motivation there is unclear to me.
- What is $\sigma^2$ in Table 2.
- What are the connections to the convergence results to Liu et al and Ismailov et al? Since you mention their condition is a special case?
- You mention in line 202 that you will experimentally observe that there are few problematic points. Where is the reference to that in the experimental section?
- There is no any further discussion of Corollaries 1 and 2.
- Based on your motivation you are not interested in measuring convergence to stationary point right? What is the motivation of considering P = $\|| \nabla f \||_2$ then?
- I think you should mention https://arxiv.org/pdf/2407.01825 and discuss their relevance to your experiments.
- Why do you mention $c_2$ and not the residual $C^K$ in the experiments?
- I am confused with your mentioned numbers about bounds on $c_2$ in the experiments. Can you please put a dashed line showing those thresholds? Or having a graph where f(x) shows how many of the iterates had $c_2 \geq x$ would be nice.
- You use $k$ and $K$ in the labels of the plots which makes it a bit confusing.
- Why do you train the ResNet with Adam in the experiment? Training with SGD makes more sense in this setting.
- Would be good to have discussion and conclusion.

---

### Official Review · Reviewer_YtLh · 2025-11-01

**Soundness:** 2
**Presentation:** 2
**Contribution:** 2
**Rating:** 2
**Confidence:** 4

**Summary:**

This paper introduces a new parametric assumption for nonconvex functions that generalizes several classes of previously considered assumptions, and studies gradient method under this assumption. Under different parameters, the assumption recovers existing assumptions such as weak quasi-convexity, aiming and $\alpha$, $\beta$ assumptions conditions. The authors establish unified convergence theorems for both deterministic and stochastic settings and validate their framework empirically on practical optimization trajectories.

**Strengths:**

1. The paper generalized previously considered assumptions and provide unified analysis of gradient method under this assumption. For different choices of $c_1, c_2, P$ they recover some existing convergence rates.

**Weaknesses:**

1. It seems that the paper generalizes some previously considered conditions but does not provide a practical applications of the new assumption.
3. Most of the rates are recovered for the case when $c_2 = 0$, which, as presented in Table 1 of the paper, are already satisfied by previous conditions.
4. Assumption 2 is characterized by the set $X$ , but the set $X$ is not defined in any of the examples. Please provide a discussion on the set $X$.
5. In Section 2 of the paper, Examples 1–3 recover known classes satisfying Assumption 2 with particular choices of $c_1, c_2, \tilde{S}, P$, and $ X $. I believe it would be beneficial to also provide the best-known rates for this class of functions from previous works.
6. The experiments do not provide enough details. The step size is not specified. Moreover, there is no connection between the introduced assumption and the problems considered in the Experiments section. It would be beneficial to connect the theoretical section of the paper with practical examples that satisfy Assumption 2.

**Questions:**

1. Do the functions $f_1, \ldots, f_5$ from Table 1 satisfy $\alpha, \beta$ condition from Islamov et al. 2024?
2. On line 202, regarding the point x^k belonging to X: could you please elaborate more on what it means and what set $X$ is/or might be? It seems that the newly introduced property of Assumption 2 is characterized by c_1, c_2, \tilde{S} and set X. Why does Table 1 not present the set $X$ for functions $f_1, \ldots, f_5$?
3. Are the any examples of the functions belonging to the new proposed class that does not satisfy previously considered conditions?

---

### Meta-Review · Area_Chair_Re7f · 2025-12-29

**Summary:**

This work introduces a new assumption (in replacement of the common convexity assumption) that guarantees global convergence of the gradient descent algorithm. The new assumption includes convexity as a special case, but some reviewers point out that it is not much more general than convexity, as it is not applicable to most deep learning models.

**Reviewer Concerns:**

No rebuttal is provided.

**Reviewer Scores:**

None is likely to change.

---

### Decision · Program_Chairs · 2026-01-26

Reject